# The Roles of Polyamines in Intestinal Development and Function in Piglets

**DOI:** 10.3390/ani14081228

**Published:** 2024-04-19

**Authors:** Bihui Tan, Dingfu Xiao, Jing Wang, Bi’e Tan

**Affiliations:** 1Key Laboratory for Quality Regulation of Livestock and Poultry Products of Hunan Province, College of Animal Science and Technology, Hunan Agricultural University, Changsha 410128, China; echotbh@163.com (B.T.); xiaodingfu2001@hunau.edu.cn (D.X.); jingwang023@hunau.edu.cn (J.W.); 2Yuelushan Laboratory, Changsha 410128, China; 3Hunan Linxi Biological Technology Co., Ltd. Expert Workstation, Changsha 410202, China

**Keywords:** piglets, intestine development, polyamine metabolism, nutritional intervention

## Abstract

**Simple Summary:**

The development and maturation of the intestine is important for piglets to maintain optimal growth and health. Polyamines play key roles in differentiation, migration, remodeling and integrity of the intestinal mucosa after injury. The extensive reviews about the roles of polyamines in intestinal development and function in piglets are presented. The nutritional intervention based on polyamine metabolism is an effective strategy to improve intestinal development and functions, especially to promote the adaption to weaning in piglets.

**Abstract:**

The gastrointestinal tract plays crucial roles in the digestion and absorption of nutrients, as well as in maintenance of a functional barrier. The development and maturation of the intestine is important for piglets to maintain optimal growth and health. Polyamines are necessary for the proliferation and growth of enterocytes, which play a key role in differentiation, migration, remodeling and integrity of the intestinal mucosa after injury. This review elaborates the development of the structure and function of the intestine of piglets during embryonic, suckling and weaning periods, the utilization and metabolism of polyamines in the intestine, as well as the role of polyamines in intestinal development and mucosal repair. The nutritional intervention to improve intestinal development and functions by modulating polyamine metabolism in piglets is also put forward. These results may help to promote the adaption to weaning in pigs and provide useful information for the development and health of piglets.

## 1. Introduction

The intestine plays a crucial role not only in digestion, absorption and metabolism of nutrients but also in selective barrier protection against harmful antigens, pathogens and toxins [1,2]. The piglets suffer abrupt changes and challenges at birth and weaning [3]. The development and maturation of the intestinal tract and immune system are important to survive these changes and achieve a high level of intestinal health. Intestinal and mucosal development is driven by accelerated epithelial cell proliferation [4,5]. The change in the expression of biologically active enzymes and transporters provides the cellular basis for intestinal physiology and immunology development [6]. The high rates of dynamic turnover and cell proliferation of intestinal epithelium are regulated by cellular polyamines, which have been recognized as the instant energy sources and the intracellular messengers regulating the growth of gastrointestinal mucosa [7]. The polyamines are small aliphatic amines and characterized by having two or more amino groups that include putrescine, spermidine and spermine spermidine (N-(3-aminopropyl)-1,4-butane diamine), spermine (N,N-bis (3-aminopropyl)-1,4-butane diamine), and putrescine (1,4-butane diamine) [7]. Polyamines are derived from foods and endogenous synthesis, and proper proportions of plant- and animal-source products provide individuals with polyamines for human health [7,8]. However, data on polyamine contents in animal feedstuffs are limited, and available data in the literature show large variations which could be due to different analytical techniques [9]. As dietary polyamines are absorbed before they reach the posterior intestine, endogenously generated polyamines are crucial for intracellular polyamine homeostasis [8]. Arginine, glutamine and proline, as the substrates for polyamine synthesis in the intestinal mucosa, have been demonstrated to stimulate intestinal epithelial cell proliferation and prevent intestinal mucosal atrophy [10]. This review presents the recent developments in the regulation of polyamines and their precursors in intestinal development and function in piglets.

## 2. Development of Structure and Function of Intestine in Piglet

### 2.1. Intestinal Organogenesis and Growth during Embryonic Period

Intestinal organogenesis begins at the early stage of embryo, when endoderm cells migrate to form a simple intestinal tube surrounded by mesenchymal cells [11]. Subsequently, the endoderm formed by undifferentiated cells transforms into villi with simple columnar epithelial cells, which in turn differentiate into intestinal epithelial cells with a microvilli structure, brush-edge hydrolases, and nutrient transporters. Proliferating cells aggregate in the area between the naive villi and invade to basal lamina to form a primitive crypt [11]. The crypt contains pluripotent stem cells, which proliferate and migrate to the top of the villus along the crypt–villus axis, differentiating into absorptive enterocyte, goblet cells, Paneth cells and endocrine cells. In addition, M cells and tuft cells also exist in the intestinal epithelium [12]. Then, the fetus begins to swallow and digest amniotic fluid, from which the fetus absorbs 10% to 14% of the nutrients and energy for its own tissue synthesis and development [13]. In addition to nutrients, amniotic fluid contains biologically active non-nutritive substances, such as growth factors and hormones, which are critical for regulating fetal gut development.

### 2.2. Intestinal Structure Development of Suckling Piglets

In order to adapt to the transition from an intrauterine growth environment to an independent extrauterine growth environment, the intestinal structure and function of neonatal piglets undergo great changes. After birth, colostrum stimulates the rapid growth of the intestinal tract, which is mainly manifested by increased intestinal DNA synthesis and protein content, decreased cell turnover rate, and accompanied by the expansion of villi and microvilli surface area [3]. The relative weight of the small intestine at 24 h after birth is twice that at birth [4]; the intestinal crypt depth and villus height is increased by 40% and 30%, respectively [14]; and the intestinal digestive area is increased by 50% [6,15]. During the first 2 to 3 days, intestinal epithelial cells absorb proteins (immunoglobulin, hormones, growth factors, etc.) from colostrum and store them in digestive vacuoles, resulting in a dramatic increase in cell size. Digestive vacuoles disappear from the proximal jejunum to the distal ileum along with the intestine maturing [16]. The intestinal surface area is increased when the mucosa folds to form plicae circulares with multiple crypt–villus units. At 1 week after birth, intestinal mucosal blood flow increases, villi length also increases, and a large number of transverse furrows appear on the villi surface gradually. However, villi become shorter and thicker, and the number and depth of transverse furrows decrease with age [16]. Wang et al. found that the jejunal microvilli were long, thin and sparse at 1 day of age, but gradually became short, thick and dense with age, while the number of intestinal goblet cells gradually increased in piglets. At 14 days of age, the villi changed from finger-like at birth to tongue-shaped, crypt depth was further deepened, and the villi surface became smooth compared with 7 days of age [17].

### 2.3. Development of Intestinal Digestion and Absorption Functions in Piglets

The expression of biologically active enzymes and transporters, which degrade and transport nutrients from the intestinal lumen across the epithelium, is essential to the digestive function of the intestine [6]. The prenatal development of digestive enzymes is an important component in the preparation for life and nutrition ex utero. The expression of brush-border enzymes is linked to the change in enterocyte generations. Among the major intestinal brush-border enzymes of piglets, only lactase and peptidase activities reach notable levels in the fetus, and they reach peak tissue-specific activities at birth or shortly after birth [6,18]. More gradual changes in intestinal enzyme activities occur after the colostrum period. The activities of lactase, sucrase, maltase and aminopeptidase in mucosal brush-border are gradually increased after 0, 6, 12 and 24 h of sucking colostrum [19]. Sucrase activity can be detected in the jejunum and increases rapidly with age, reaching 55-fold levels at 9 weeks [4]. The maximal enzyme activity (Vmax) and enzyme affinity (Km) of sucrase increased from days 8 to 70 of age [20]. For aminopeptidase N, Vmax was low in the suckling and high in the post-weaning stage; Km was the highest in the weaning stage, intermediate in the adult and the lowest in the suckling stage [20]. Maltase activity, which is low in intestinal mucosa at birth, increases gradually from birth to 6 or 8 weeks of age [19]. The intestinal ion and nutrient transports of piglets start to develop during early fetal life in order to absorb the amniotic fluid components. Once epithelial cells in the small intestine differentiate into columnar cells during pregnancy, brush-border transporters such as sodium/glucose co-transporter 1 (SGLT1) are expressed. Neonates maintain a high uptake activity of intestinal apical SGLT1 by abundantly expressing SGLT1 protein in the epithelia and on the apical membrane along the entire crypt-villus axis [21]. An age-related decrease in ion and nutrient fluxes takes place in the immediate postnatal period [22], for example, the jejunal apical amino acid transporters (B^0^AT1, ASCT2 and EAAC1) protein abundances were dramatically decreased during the postnatal development [23]. However, the total absorption capacity increases with the increase of intestinal mass. Therefore, the decreases in carrier-mediated amino acid absorption within 24 h after birth are caused not by the loss of transporters but by the rapid increases in tissue mass that effectively ‘dilute’ the transporters [24]. Significantly, the declines of carrier-mediated amino acid and glucose transport also coincide with the postnatal replacement of the fetal enterocytes, leading to a redistribution of transport functions along the crypt–villus axis [25].

### 2.4. Development of Intestinal Immune System in Piglets

The intestinal immune system of piglets has been evolved to maintain the integrity of the epithelial surface and provide a barrier against pathogens, antigens and toxins [3]. The development and maturation of the intestinal mucosal immune system is initiated in the intrauterine environment. Maternal microbes may be involved in the establishment of intestinal immune and barrier functions in offspring by vertically transmitted maternal milk and intestinal microbes [26]. Maternal microbiota shapes the immune system of the offspring; gestational colonization increases intestinal group 3 innate lymphoid cells and F4/80^+^CD11c^+^ mononuclear cells in the offspring [27]. Then, birth and weaning are major challenges to the development of the mucosal immune system, when it must adapt to gastrointestinal microbial colonization, breast milk and feed antigens [28,29,30]. The induction of intestinal immune reactions starts with antigen presentation by professional antigen presenting cells of Peyer’s patches and mesenteric lymph nodes [28]. There are few lymphocytes in the intestinal epithelium or lamina propria of neonatal piglets at birth. Then, the lymphocytes are rapidly accumulated by antigen stimulation in the first 2 weeks after birth [30]. CD4^+^ cells are observed between 14 and 28 days of age, but CD8^+^ cells do not appear until 35 days of age in the intestinal mucosa [28]. By 7 weeks of age, the architecture of the intestine is comparable to that of the mature animal [30]. However, piglets are weaned at 3~4 weeks of age in commercial practice, and this results in an increased susceptibility to bacterial infection and acute diarrhea and high mortality rates [3]. The obvious increase of the numbers of CD2^+^ leucocytes was observed in the intestine, and piglets showed reduced ability to react to the lymphocyte mitogen phytohemagglutinin after weaning. With the reduction of interleukin (IL)-2 secreted from systemic T cells, production of specific antibodies was also reduced [31,32].

### 2.5. Morphological and Functional Changes in the Intestine of Weaned Piglets

The intestine of neonatal piglets is regarded as anatomically and functionally immature as to be hypersensitive to weaning [3,33]. Weaning stress results in morphological and functional changes in the small intestine, such as villous shortening, crypt elongation and reduction of activities of digestive enzymes and nutritional transports, which lead to a depressed digestive capacity and defense function of the intestine [5,18]. The atrophy of small intestinal villi is mainly due to the slow rate of intestinal cell proliferation and the shedding of a large number of mature enterocytes at the top of the villi. The decrease in intestinal villi height results in a shortened migration distance (EMD) from the base of the crypt to the top of the villi. The shortening of EMD accelerates the migration of enterocytes, which is 1.86, 3.42 and 2.60 times faster at days 2, 4 and 8 than at day 0 of post-weaning (PW) [3]. The most severe damage of the intestinal mucosal barrier occurs between 2~5 d PW [17,34]. The small intestine and its mucosa lose 20%~30% of relative weight during the first 2 d PW, and the villous height may be reduced to 75% of pre-weaning values within 24 h of weaning at 21 days of age [35]. Earlier weaning at 14 days of age results in significantly lower gene expression levels of E-cadherin and Occludin and protein expression of Occludin and zona occludens (ZO)-1 in the jejunum on days 3 and 5 PW [17]. Generally, with the reduction in villous height and increases in crypt depth, the specific activity of brush-border enzymes lactase and sucrase declines sharply and reaches minimum values 4~5 days after weaning. Sucrase-, isomaltase- and lactase-specific activities decline by at least 15% on day 5 after weaning in piglets weaned at 28 d of age, and enteric infections further depress intestinal enzyme activities [6,20]. However, the effects of weaning on intestinal enzyme activities appear to be dependent on age at weaning [6]. An early weaning at 7 d of age induces no alteration or even an increase in sucrase, maltase, and peptidase activities during the immediate post-weaning period [6]. Such age-dependent variations in brush-border enzyme activities may be due to variations in cellular renewal rate and protein synthesis [6].

Intestinal mucosa is capable of rapid repair after injury. The intestinal mucosal barrier can be recovered at 5~10 days PW [35]. Hu et al. [34] found that villus height and crypt depth can return to a pre-weaning level on 14 d PW. The repair of intestinal injury in piglets is a very complex process, including reconstruction of epithelial cells, cell proliferation, apoptosis, angiogenesis and mucosal remodeling. Crypt cells are continuously proliferated and differentiated, new cells are migrated along the crypt–villus axis, during which they gradually become mature and fully functional villus cells. Rapid reconstruction of mucosa depends on intestinal polyamines, which can be synthesized de novo or ingested from the intestinal lumen [36].

## 3. The Role of Polyamines in Intestinal Development and Mucosal Repair in Piglets

### 3.1. Metabolism and Utilization of Polyamines in the Intestine

Polyamines in the intestinal lumen are mainly derived from dietary intake, microbial synthesis and broken cells shed from the top of villi [37,38]. For newborn piglets, bacterial fermentation of protein is the main source of polyamines in the colon [39]. The concentration of polyamines in chyme varies by feed composition or feeding stage. Higher polyamine levels in the lumen of the proximal colon are observed in pigs fed a casein diet compared with a soy diet [40]. However, putrescine and cadaverine in the duodenal and colonic luminal chyme were over 90% free, with only 10% or less bound to protein [41]. Polyamines in chyme are rapidly absorbed by enterocytes, showing the highest polyamine concentrations after feeding, with a significant decrease 2 h later in the lumen [42]. In an isotope marker perfusion test, 61%~79% of radioactive substances can be detected in venous blood 10 min after ^14^C-labeled putrescine, spermine and spermidine perfusion [43]. When de novo synthesis of cellular polyamines is impaired or there is an increasing need for polyamines, the utilization of exogenous polyamines is increased [44].

Polyamines are absorbed in the duodenum and anterior jejunum. It has been proposed that polyamines enter cells by passive diffusion, paracellular pathway, endocytosis and so on [38,42,45]. Polyamines are transported into cells by membrane transporters and then accumulated into polyamine vesicles, which are driven by vesicles ATPase and pH gradient and involved in proton exchanges [45]. The uptake of polyamines is also associated with heparin sulfate proteoglycan and glypican on the surface of cells. Putrescine and acetyl polyamines can exit from the cell through amino acid transporter SLC3A2, during which process Arg is exchanged with putrescine [38].

Homeostasis of polyamines can be regulated by their own feedback mechanisms, including de novo synthesis, uptake and catabolism. The interconversion of various forms of polyamines via acetylation and oxidation controls polyamine turnover according to physiological needs. Polyamines are partly metabolized in the intestine by spermine oxidase (SPMO), spermidine oxidase (SPDO) and spermine/spermidine-N-acetyltransferase (SSAT) [46]. Polyamine degradation products, acetaldehyde and hydrogen peroxide are cytotoxic. Under normal conditions, intracellular activities of catalase, glutathione peroxidase and acetaldehyde dehydrogenase are high enough to prevent the accumulation of cytotoxic oxides [8]. Antizyme (AZ) protein can bind to ornithine decarboxylase (ODC) and promote the degradation of ODC, thus inhibiting the endogenous synthesis of polyamines [42]. And the activity of AZ can be feedback-regulated by polyamines and then it down-regulates polyamine transport [36]. Diamine oxidase (DAO) oxidizes putrescine, as well as the amino group of spermine and spermidine, and is responsible for the elimination of excess polyamines in the intestinal and systemic circulation [42,47].

### 3.2. Polyamines Promote Intestinal Development and Maturation

Polyamines are important regulatory factors of angiogenesis, early embryogenesis, placental trophoblast growth and embryo development in mammals [48]. In the placenta of sows, polyamines are mainly synthesized by Pro through peroxidase (POX), ornithine aminotransferase (OAT) and ODC because of the lack of arginine degradation by arginase [48,49]. The rate and concentration of polyamine synthesis in the placenta of sows peak at day 40 of gestation, and decrease significantly at day 60 to 90 of gestation. The concentrations of Pro in amniotic fluid are also increased by 130% at day 20 to 30 of gestation, followed by a decrease in the second trimester, then a significant increase at day 90 to 110 [50]. Maternal nutrients during gestation contribute significantly to the polyamine pool in the fetal intestinal tract. The fetus uptake of polyamines from amniotic fluid supports the differentiation and proliferation of enterocytes in the embryonic stage, which directly affects intestinal growth and development after birth [51]. In IUGR fetuses, lower abundances of intestinal transporters result in the decreasing uptake of polyamines from amniotic fluid or the umbilical vein [52]. Accumulation of polyamines during early gestation may promote the development of the fetal intestine during late gestation. The previous study shows that dietary supplementation with Pro in pregnant sows increases protein and DNA concentrations in the small intestine of fetal pigs [53].

During the suckling period, sow colostrum and milk provide polyamines for piglets to modulate intestinal maturation [54]. The spermidine concentration in sow milk peaks at 7 weeks, coinciding with a rapid increase of intestine mucosal RNA content and maltase activity in suckling piglets [55]. Oral administration of putrescine or proline during the suckling period can prompt the maturation of the intestine manifested by increasing small intestinal villus height, the percentage of proliferating cell nuclear antigen (PCNA) positive cells, and alkaline phosphatase (AKP) activity in the jejunal mucosa [56]. Putrescine can convert to succinate that provide a direct energy source to promote intestinal epithelial turnover [57]. Daily supplementation with spermine or spermidine during suckling has been reported to enhance the intestinal maturation, including immune cells that increase the percentage of TCRαβ^+^, CD4^+^, CD5^+^ and CD54^+^ intraepithelial lymphocytes [58].

The change of the polyamine metabolism pattern is accompanied by the maturation of tissues and organs. The gradual decreases of ODC and S-adenosylmethionine decarboxylase (AdoMetDC) activities as well as the concentrations of putrescine, spermine and spermidine indicate the transition to maturation [59]. With the differentiation of enterocytes and ODC activity, the concentration and transporters of putrescine are significantly increased at the early proliferation stage [38]. The effects of polyamines on promoting enterocyte proliferation are involved in multiple possible mechanisms. The physiological processes of cell division, protein synthesis and de novo synthesis of mRNA and DNA are dependent on intracellular polyamines [40]. Depletion of intracellular polyamines inhibits protein production and the synthesis of RNA and DNA. Polyamines have been reported to regulate the expression of nuclear transcription factors including c-Fos, c-Jun and c-Myc, which regulate the cell cycle and participate in cell growth and development [58]. Polyamines may also promote the proliferation of intestinal stem cells by reducing Smad signaling through modulating the transforming growth factor (TGF)-pathway [60].

### 3.3. Polyamines Improve Intestinal Mucosal Damage Repair

The various adaptive changes after weaning are, substantially, the intestinal maturation [61]. The essence of intestinal maturation is intestinal reconstruction, i.e., the gradual replacement of original intestinal cells with newly differentiated and mature intestinal cells [62]. The repair of intestinal mucosal injury mainly depends on the continuous proliferation and differentiation of cells located in the crypt and the migration of epithelial cells, so as to restore epithelial integrity and maintain mucosal barrier function [46]. It has been shown that polyamines are essential for the proliferation, differentiation, migration and reconstruction of intestinal cells in weaned piglets [56]. Intracellular polyamines are closely related to the repair process of intestinal mucosa [63]. The concentrations of spermine and spermidine in the jejunum and ileum of piglets are increased from day 1 to 3 after weaning, and the gene and protein expression of ODC, the first rate-limiting enzyme for polyamine synthesis, are significantly increased on day 3 PW [64]. Supplementation with putrescine and proline during the suckling period can promote epithelial restitution after weaning. The polyamine-mediated mucosa restitution is partially due to increased levels of voltage-gated potassium channel (Kv) activity, and the expression levels of Kv1.5 mRNA and Kv1.1 protein in ileal mucosa are increased in response to proline administration [56]. Activation of potassium channels is related to the growth or differentiation of a variety of cells. Polyamines induce hyperpolarization of cell membranes and promote Ca^2+^ influx by increasing the expression of K^+^ channels [65]. At the same time, the ratio between stromal sympathetic molecule 1 (STIM1) and STIM2 is changed by regulating the transient potential receptor channel 1 (TRPC1) to release Ca^2+^ [46,63]. During mucosal remodeling, an increase of intracellular Ca^2+^ concentration regulates certain proteins to promote cell migration, such as the up-regulation of phospholipase C and the small G protein RhoA [66]. Mucosal reconstruction of more serious damage involves angiogenesis and fibroblast proliferation [67]. Polyamines can significantly increase gastrointestinal blood flow and accelerate ulcer healing [68], but the regulatory mechanism has not been fully elucidated.

### 3.4. Polyamines Maintain Intestinal Barrier Function

More and more studies have proven that polyamines are necessary for maintaining intestinal barrier function. Polyamines increase the expression of molecules involved in adherent junctions, thus promoting intestinal barrier function [36]. Conversely, polyamine depletion inhibits the expression of Occludin, Claudin, E-cadherin and ZO-1, thereby partially increasing the permeability of epithelial cells [67]. It has been demonstrated that the reduction of polyamine levels is related to the dysfunction of the epithelial barrier, and exogenous polyamines can improve the dysfunction of intestinal barrier function in some cell and animal models [69,70]. In IEC-6 cells, the absence of polyamines down-regulated the expression of tight junction proteins 1 and 2, Occludin, Claudin 2 and 3 (CLDN2,3) and E-cadherin [71]. In piglets, oral supplementation with putrescine and proline during the suckling period increases the protein expression of ZO-1, Occludin and Claudin-3 in the jejunum after weaning [56].

Polyamines regulate tight junction protein expression through different cellular signaling pathways. A decrease of polyamine concentration leads to the increases of JunD expression. JunD can increase the interaction between the C-AMP-binding-site protein in the proximal promoter region and RNA-binding protein in the 3′ non-coding region of ZO-1, thus reducing the protein expression of ZO-1 in Caco-2 cells [72]. The depletion of polyamines also reduces the expression of checkpoint kinase 2 (Chk2) and phosphorylation HuR protein, which inhibits the translation of Occludin mRNA in IEC-6 cells. By adding exogenous polyamines, the inhibition of Chk2 can be eliminated and the phosphorylation of HuR proteins can be restored, thus promoting the translation and expression of Occludin [69,73]. In addition, polyamines can also promote the translation of E-cadherin mRNA and increase the stability of this protein in epithelial cell lines [74].

## 4. The Nutritional Intervention for Polyamine Metabolism and Intestinal Function in Piglets

### 4.1. Dietary Polyamines Supplementation

Polyamines are present in all types of foods in a wide range of concentrations [8]. Available data about polyamine content in feedstuffs are limited and variable. Natural differences in polyamine content among foods reflect differences in the metabolism of polyamines among different species and even among different varieties of the same species [75]. For feeds, the raw materials used (e.g., their sources, freshness, microbial contamination) and processing technologies for ingredient production can also lead to different polyamine contents [75]. And the variations in published values might be attributable to different analytical techniques used in various laboratories [8,9]. Hou et al. determined the content of polyamines in major staple plant foods consumed by humans using high-performance liquid chromatography (HPLC) methods and found that total polyamines were most abundant in corn grains, followed by soybeans, sweet potatoes, pistachio nuts, potatoes, peanuts, wheat flour and white rice in descending order on the dry matter basis [9]. The main polyamine in plant-based products is spermidine, whereas spermine content is generally higher in animal-derived foods. Among plant-derived foods, vegetables and fruits have the highest putrescine content, and cereals, legumes and soybean derivatives have the highest content of spermidine and spermine. In animal-derived foods, meat and derivatives have the highest polyamine content, with the exception of some cheeses [8].

Although there are no recommendations for polyamine daily intake, it is known that polyamine requirements are high for piglets in the neonatal period with rapid growth [57]. Maternal milk polyamine levels are higher during the first week of lactation, which meets the polyamine needs for rapid growth and gut maturation in piglets. A milk formula supplemented with polyamines at maternal milk physiologic doses (5 nmol/mL of spermine and 20 nmol/mL of spermidine) enhances gut growth and maturation in neonatal piglets [76]. Oral administration with 5 mg/kg BW putrescine twice daily from day 1 to weaning at 14 d of age improves mucosal proliferation, intestinal morphology, as well as tight junction and potassium channel protein expressions in early-weaned piglets [56].

### 4.2. Improve Endogenous Polyamine Synthesis by Their Precursors’ Administration

Because dietary polyamines are readily absorbed by enterocytes of the small intestine, endogenously generated polyamines are crucial for intracellular polyamine homeostasis [8]. The administration of precursors has been extensively studied to improve polyamine metabolism and intestinal functions in piglets [56,77]. In mammals and microbes, arginine, agmatine, proline, glutamine, glutamate, ornithine and methionine are substrates for polyamine synthesis. Arginine has been proven to promote the differentiation and proliferation of enterocytes, reduce cell apoptosis, prevent intestinal villi atrophy and so on [78,79]. However, arginase activity is absent from placentae of rapidly growing porcine and enterocytes of suckling piglets [80]. Therefore, the beneficial effects of arginine on intestinal development and function in piglets may be involved in the nitric oxide (NO) pathway rather than polyamine pathway. Inhibition of inducible NO synthase by L-NG-nitroarginine methyl ester abrogated the effect of L- arginine on preserving intestinal integrity under heat stress [81]. On the other hand, the expression of POX (oxidize proline to ornithine) is much higher in the intestinal tract of newborn piglets. Therefore, proline is the main source of polyamines in the intestine of piglets [48,49]. Wu et al. [48] have demonstrated that the increase of plasma cortisol concentration stimulates intestinal ODC activity and polyamine synthesis from proline in suckling and weanling piglets. Oral administration of proline can increase the concentrations of spermidine and spermine in the ileal mucosa and ODC protein expression in the intestine [56,64].

### 4.3. Intervene Polyamine Metabolism via Manipulating Intestinal Microbiota 

Most polyamine in the posterior intestine are derived from gut microbiota, which are critical for the polyamine pool homeostasis [39]. In the colonic lumen, putrescine is produced from arginine via the collective biosynthetic pathway, consisting of multiple pathways catalyzed by enzymes derived from multiple bacterial species [41]. Spermine can be synthetized through endogenous polyamine metabolism mediated by *Actinobacteria*, *Firmicutes*, *Proteobacteria* and *Bacteroidetes* [82]. *Lactobacillus curvatus* KP3-4 from the traditional Japanese fermented food “kabazushi” can produce high levels of polyamines, which has been shown to produce ∼90 m M putrescine in the presence of 750 m M ornithine [83]. Therefore, nutritional interventions for polyamine metabolism via modulating intestinal microbiota are alternative strategies [84,85,86,87]. *Lactobacillus curvatus* KP3-4 has been demonstrated to increase the concentrations of putrescine and spermidine in the intestinal lumen of colonized mice [83,84]. Supplementation of probiotic strain *Bifidobacterium animalis* subsp. *lactis* LKM512 has also been shown to increase the colonic polyamine concentration in mice [86]. However, this increase of polyamine content likely results from the activation of polyamine biosynthesis by indigenous gut microbiota because *Bifidobacterium* spp. does not have homologs of enzymes involved in polyamine biosynthesis [86].

In addition, the presence of fermentable carbohydrates may contribute to the increase in bacterial polyamine synthesis and maintaining the integrity of the gut mucosa [39]. The intake of fructooligosaccharide (FOS) increased the proportion of *bifidobacteria* and *lactobacilli*, as well as polyamine concentrations in the cecal content of neonatal piglets. *Bifidobacterium* and *lactobacilli* are able to produce polyamines, but they seem not to be the main microbial producers of polyamines in the cecum of piglets fed FOS [88]. Some phytochemicals have been reported to enhance the synthesis of functional amino acids (e.g., arginine, glutamine and proline) and polyamines, which may be involved in intestinal microbiota [89,90]. For example, *Yucca schidigera* extract can stimulate arginine degradation for polyamine synthesis in pig enterocytes [89]. *Sijunzi* decoction also plays a prophylactic effect on mucosal injury by increasing the polyamine content in the small intestine [91]. *Atractylodes macrocephala Koidz* significantly stimulates the migration of intestinal epithelial cells through the polyamine-Kv1.1 channel signaling pathway and promotes the healing of intestinal injury [92].

## 5. Conclusions

As reviewed above, polyamines play critical roles in the intestinal development and functions of suckling and weaning piglets (Figure 1). Based on the research progress of polyamine utilization and metabolism in pigs, some nutritional interventions to improve intestinal development and functions have been developed. The regulation of polyamine metabolism targeting intestinal microbiota could be an effective strategy to improve intestinal development and function. Future studies about the mechanisms of host–microbiota interactions on polyamine metabolism and the identification of specific microbiota for polyamine synthesis could be investigated in order to provide more useful information for the regulation of development and health in piglets.

## Figures and Tables

**Figure 1 animals-14-01228-f001:**
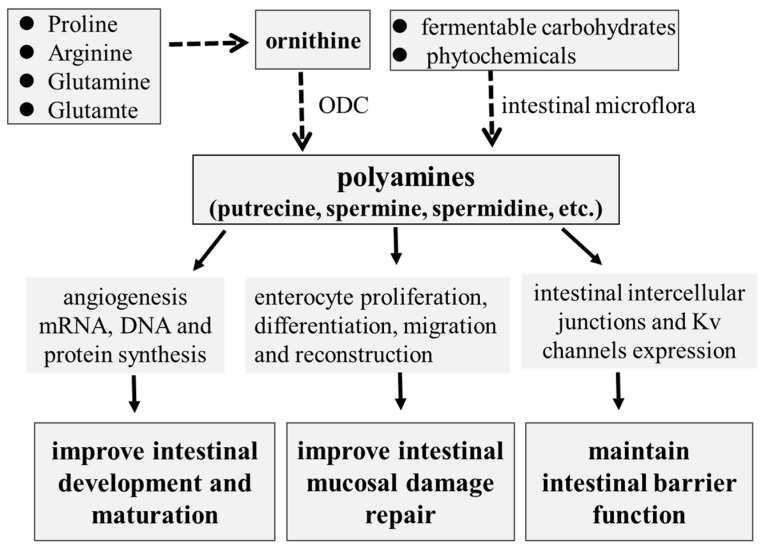
The role of polyamines in intestinal development and mucosal repair in piglets.

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
