# Peer review of "The Roles of Polyamines in Intestinal Development and Function in Piglets"

_animals, 2024, doi:10.3390/ani14081228_

Round 1
Reviewer 1 Report
Comments and Suggestions for Authors
The paper need be revised for reconsider for publication, some concerned must be answered,
1. The definition of polyamines, what are they? Is there any feedstuff contain, and how to determine it?
2. Where did they come from, precursor substances for them?
3. Is there a minimum and maximum requirement for piglets?
4. How polyamines influence intestinal development and function
Besides,
Line 183
concentrations of polyamines in digesta are putrecine and cadaver, which about 90% of
putrecine and cadaver need checked
Line 210
cytotoxic oxides [45]. Antizyme (AZ) protein can bind to ODC and promote the…
what is ODC
Author Response
- The definition of polyamines, what are they? Is there any feedstuff contain, and how to determine it?
We added the definition (Line 40-44). For feedstuff contain, we added a section “Dietary polyamines supplementation” (Line 320-347). We also refer the determination using high-performance liquid chromatography without more detail about the method.
- Where did they come from, precursor substances for them?
Polyamines are derived from foods and endogenous synthesis. In mammals and microbes, arginine, agmatine, proline, glutamine, glutamate, ornithine, and methionine are substrates for polyamine synthesis. We added this description. Section 4.1 and 4.2.
- Is there a minimum and maximum requirement for piglets?
There are currently no official recommendations for daily polyamines intake. But it is known that polyamine requirements are high for piglet in the neonatal period with rapid growth. We added this description. (Line 339-347)
- How polyamines influence intestinal development and function
Besides, Line 183 concentrations of polyamines in digesta are putrecine and cadaver, which about 90% of putrecine and cadaver need checked
Thanks. We revised it. (Line 193-194)
Line 210 cytotoxic oxides [45]. Antizyme (AZ) protein can bind to ODC and promote the… what is ODC
We have added the full name “ornithine decarboxylase” for ODC. (Line 217)
Reviewer 2 Report
Comments and Suggestions for Authors
The review emphasizes the role of the gastrointestinal tract in nutrient absorption and barrier function maintenance, particularly crucial for the optimal growth and health of piglets. The authors explore and elucidate the significance of polyamines in facilitating enterocyte proliferation and intestinal mucosal integrity, during developmental stages and mucosal repair.
General comments - Although the topic is interesting, more details must be included in the review, as most topics are presented in just one paragraph. It would strengthen the review to include more insights into the functional implications of these changes. How do alterations in villus morphology, for example, relate to nutrient absorption and overall intestinal health?
Also, incorporating transitional phrases between sentences and paragraphs could improve the flow of information and enhance readability. Paragraphs are long and sometimes hard to follow.
Line 70 - change "increased' to "increases"
Line 73 - 78 - Rewrite the sentence. I couldn't understand the statements.
Line 152 - Define PW. It is defined in line 153
Line 272 - Provide brief explanations for terms like "Kv channel activity" or "STIM1/STIM2 ratio" that could help readers understand the significance of these pathways. This can be used through the review.
Subtitle 3.4 - General comment - Be specific when in vitro or in vivo studies
Line 308 - Provide specific examples or case studies showing the efficacy of different nutritional interventions, such as fermentable carbohydrates or phytochemical supplementation.
Figure 1 - what age range do you attribute to the piglets or young piglets? It would call for more attention to the review if you could add more details.
Author Response
General comments - Although the topic is interesting, more details must be included in the review, as most topics are presented in just one paragraph. It would strengthen the review to include more insights into the functional implications of these changes. How do alterations in villus morphology, for example, relate to nutrient absorption and overall intestinal health?
Also, incorporating transitional phrases between sentences and paragraphs could improve the flow of information and enhance readability. Paragraphs are long and sometimes hard to follow.
Thanks. We try to our best to revise some.
Line 70 - change "increased' to "increases"
Thanks. Change it to “is increased” in agreement with the previous sentence. (Line 79)
Line 73 - 78 - Rewrite the sentence. I couldn't understand the statements.
Thanks. We rewrite this sentence. See Line 82-86.
Line 152 - Define PW. It is defined in line 153
Thanks. We revised it.
Line 272 - Provide brief explanations for terms like "Kv channel activity" or "STIM1/STIM2 ratio" that could help readers understand the significance of these pathways. This can be used through the review.
Thanks. We added it.
Subtitle 3.4 - General comment - Be specific when in vitro or in vivo studies
Thanks. We added it. (Line 303-317)
Line 308 - Provide specific examples or case studies showing the efficacy of different nutritional interventions, such as fermentable carbohydrates or phytochemical supplementation.
Thanks. We rewrite this section. Section 4.1, 4.2. and 4.3.
Figure 1 - what age range do you attribute to the piglets or young piglets? It would call for more attention to the review if you could add more details.
Thanks. We change “young piglets” to “piglets” that include suckling piglets and weaning piglets.
Reviewer 3 Report
Comments and Suggestions for Authors
The author reviewed the functions of polyamine in intestine development.
Unlike previous sections, section 2.4 is difficult to follow. For example, line 126 states, “Neonatal piglets have not active mucosal immune system due to hormones.” This section requires rework.
Line 119: What is the difference between the physical and immunological of the immune system?
Line 130: Colonized is not the preferred terminology for expressing cells.
Line 161: “ at least 15% one” should it be “on”?
Line 234: Does polyamine serve as a source of nutrients or function as a stimulator for these cell types?
Line 281: at line 207, authors discuss the polyamine degrade into acetaldehyde and hydrogen peroxide by SPMO, SPDO and SSAT. Is there any evident suggests that polyamine can be degraded into nitro oxide?
Line 327: the polyamine production from commensal microbes is intriguing and can add more to this article.
Author Response
Unlike previous sections, section 2.4 is difficult to follow. For example, line 126 states, “Neonatal piglets have not active mucosal immune system due to hormones.” This section requires rework.
Thanks. We rewrite this section.
Line 119: What is the difference between the physical and immunological of the immune system?
We deleted “physical and immunological”
Line 130: Colonized is not the preferred terminology for expressing cells.
Thanks. We rewrite this section. Line 134-143
Line 161: “ at least 15% one” should it be “on”?
Thanks. We revised it.
Line 234: Does polyamine serve as a source of nutrients or function as a stimulator for these cell types?
In enterocytes, polyamines serve as the instant energy sources and the intracellular messengers regulating the growth of mucosa. Line 37-40
Line 281: at line 207, authors discuss the polyamine degrade into acetaldehyde and hydrogen peroxide by SPMO, SPDO and SSAT. Is there any evident suggests that polyamine can be degraded into nitro oxide?
In animals, arginine is converted to nitric oxide (NO) via NO synthase and to ornithine via arginase. Ornithine decarboxylase (ODC) catalyzes the conversion of ornithine to polyamines.
Line 327: the polyamine production from commensal microbes is intriguing and can add more to this article.
Thanks. We added some. Line 369-384.
Round 2
Reviewer 1 Report
Comments and Suggestions for Authors
No comments